# Diagnostic Significance of FNAB miRNA Expression in Papillary Thyroid Carcinoma

**DOI:** 10.3390/diagnostics12061384

**Published:** 2022-06-03

**Authors:** Romena Laukienė, Laima Ambrozaityte, Loreta Cimbalistienė, Algirdas Utkus, Algirdas Edvardas Tamosiunas

**Affiliations:** 1Department of Human and Medical Genetics, Institute of Biomedical Sciences, Faculty of Medicine, Vilnius University, Santariskiu 2, 08661 Vilnius, Lithuania; laima.ambrozaityte@mf.vu.lt (L.A.); loreta.cimbalistiene@mf.vu.lt (L.C.); algirdas.utkus@mf.vu.lt (A.U.); 2Department of Radiology, Nuclear Medicine and Medical Physis, Institute of Biomedical Sciences, Faculty of Medicine, Vilnius University, Santariskiu 2, 08661 Vilnius, Lithuania; algirdas.tamosiunas@mf.vu.lt

**Keywords:** papillary thyroid carcinoma, miR125A, miR146B, miR221, miR4324, fine needle aspiration biopsy

## Abstract

The aim of the study was to evaluate the diagnostic utility of specific miRNAs in the preoperative assessment of thyroid nodules. One hundred and sixty thyroid fine needle aspiration biopsy (FNAB) samples with suspected thyroid carcinoma were collected. To detect the levels of miRNA expression in FNAB, next generation small RNA sequencing was performed in 60 samples. Based on the results obtained, three miRNAs (miR125A, miR200B, miR4324) were selected for further analysis. Based on the most frequently reported miRNAs in the literature associated with thyroid papillary carcinoma (PTC), two more miRNA (miR146B, miR221) were selected for further validation, using real-time reverse transcriptase polymerase chain reaction (RT-PCR) in 36 benign and 64 PTC samples. Expression of miR125A, miR146B, miR221, and miR4324 was significantly higher in patients with PTC compared with benign thyroid nodules (*p* ˂ 0.05). miR125A and miR4324 were also significantly more highly expressed in patients with extrathyroidal tumor extension compared to those without extrathyroidal PTC extension (*p* < 0.001). We also found a significantly higher expression of miR221 (*p* = 0.043) in patients with multifocal carcinomas compared to patients with single foci carcinomas. This prospective study showed that the expression analysis of four miRNAs (miR125A, miR146B, miR221, and miR4324) improve accuracy of FNAB, which could allow a better pre-operative diagnostic and prognostic assessment of thyroid malignancies.

## 1. Introduction

The prevalence of nodal disease in the general population is 50–70% [1]. Advances in ultrasound imaging have improved the early detection of thyroid cancer, but this also raises the possibility of overdiagnosis, which leads to unnecessarily aggressive surgery and associated postoperative complications that impair patients’ quality of life [2]. International professional organizations, including the European Thyroid Association (ETA) [3], the American Thyroid Association (ATA) [4] and the American College of Radiology [5], have developed risk stratification criteria for thyroid nodules, which are used to guide ultrasound findings and treatment [6]. Usually fine needle aspiration biopsy (FNAB) results are reported according to the Bethesda Thyroid Cytopathology System, with a high diagnostic specificity [7]. Nevertheless, 10–20% of thyroid nodules remain indeterminate after FNAB [8]. 

Papillary thyroid carcinoma (PTC) is one of the most frequently occurring forms of endocrine malignancy, with an increasing rate of incidence over the last three decades [9,10]. Generally, PTC has an excellent prognosis with a relatively low mortality rate, but a small portion of PTC patients suffer from an aggressive form of the disease with tumor invasion and metastasis. In clinical practice, clinicians are faced with the challenge of balancing treatment tactics so that patients at lower risk of thyroid cancer are not treated too aggressively. At the same time, they need to identify patients who have more advanced disease or are at high risk and need a more aggressive treatment approach [11]. Therefore, it is important to look for new biomarkers that can more accurately diagnose thyroid carcinoma and assess the spread of disease before the surgery.

MicroRNAs are non-coding small RNA molecules consisting of 18–24 nucleotide sequences. They underlie many oncogenesis-related processes such as proliferation, cell division cycle control, apoptosis, differentiation and migration [12,13]. Analysis of miRNAs expression in FNAB samples may improve perioperative decision making for patients with PTC, as appears to be the case in other types and locations of cancers, e.g., miR21 in breast carcinoma [14], miR31 in colon carcinoma [15] and miR1207-5p in gastric carcinoma [16].

The aim of the study was to determine and compare miRNAs expression of PTC and benign thyroid nodules to determine the diagnostic utility of the detection of specific miRNAs in the preoperative assessment of thyroid nodules.

## 2. Materials and Methods

### 2.1. Subjects

The study was approved by Vilnius Regional Bioethics Committee (Lithuania, approval No. 158200-17-905-414; 17 March 2017). All samples were collected upon obtaining written informed consent from patients and prior approval for this study as per the Declaration of Helsinki. From May 2017 to April 2021, all patients who underwent a diagnostic FNAB of a thyroid nodule at Vilnius University Hospital Santaros Klinikos and were suspected of having thyroid carcinoma (Bethesda cytology grades III–VI) were offered the opportunity to participate in the study [7]. All patients with cytologically suspected (Bethesda cytopathology grades III–VI) and postoperatively histologically confirmed papillary thyroid carcinoma or benign thyroid nodules were included. Post-operative confirmation of other histological diagnoses was an exclusion criterion. A total of 160 patients were enrolled and subsequently underwent thyroid surgery. The first 60 patients formed the screening cohort, the next 100 patients the validation cohort.

The 60 patients in the screening cohort were tested for miRNA expression by next-generation sequencing. These subjects were divided into three groups according to the post-operative final histological result: 12 subjects with benign thyroid nodules (BTN), 33 subjects with malignant nodules without neck lymph node metastases (LNM), 15 subjects with malignant nodules with LNM.

Based on the results of the sequencing, the three miRNAs that were most divergent between the groups—miR125A, miR200B, miR4324—were selected for further analysis. Following the literature review, two more miRNAs, miR146B and miR221, which have been mainly described for differentiating both PTC with benign thyroid disease and more aggressive forms of PTC [17,18,19,20,21], were selected for further investigation.

Real-time reverse transcription polymerase chain reaction (RT-PCR) was performed to investigate the expression of selected miRNAs in 100 patients from the validation cohort. These patients were also divided into analogous groups according to the final histological result: 36 benign nodules, 34 malignant nodules without cervical LNM and 30 malignant nodules with LNM. 

FNAB under ultrasound control of thyroid nodules was performed according to our hospital’s locally approved diagnostic protocol: nodules were aspirated two to three times with a 21-gauge needle. Part of the thyroid tissue material obtained from the FNAB was spread evenly in a thin layer on the slide. Liquid-based samples of the rest of the tissue material were made with the needle rinsed in ThinPrep solution. In addition, part of the liquid-based biopsy samples was immediately stored at −80 °C until RNA extraction.

### 2.2. Total RNA Isolation

Total RNA was purified from thyroid nodule fine needle aspirates using the commercial Quick-DNA/RNA™ Microprep Plus Kit (Zymo Research, Irvine, CA, USA). RNA quantification was performed using a Nanodrop spectrophotometer (Thermo Fisher Scientific, Waltham, MA, USA). Isolated total RNA samples were stored at −80 °C. 

### 2.3. Small RNA Sequencing

MiRNA profiling was performed by small RNA sequencing. Small RNA libraries were prepared using Ion Total RNA-Seq Kit v2 for Small RNA Libraries (Thermo Fisher Scientific, USA) according to the manufacturer’s protocol using 1 μg of RNA per sample. This step was followed by RNA 3′ adapter ligation, RNA 5′ adapter ligation, cDNA synthesis, and PCR amplification using unique barcode sequences for each sample. Multiplexed libraries were then sequenced using the Ion Torrent PGM (Thermo Fisher Scientific, USA) platform. Differential expression analyses were performed in Partek^®^ Flow^®^ (Partek incorporated, Chesterfield, MO, USA). MiRNAs with a mean number of reads greater than 20 in the sample were included in the analysis, with a statistically significant fold change in expression increasing/decreasing >2, *p* < 0.05.

### 2.4. RT-PCR

MiRNA expression levels were measured by quantitative reverse transcriptase-polymerase chain reaction (RT-PCR) using TaqMan^®^ Advanced miRNA Assays (Applied Biosystems, Waltham, MA, USA) kits. The cDNA was synthesized from the purified total RNA using the TaqMan miRNA Reverse Transcription Kit according to the manufacturer’s protocol. The synthesized cDNA was amplified using TaqMan miRNA probes. Three PCR reactions were performed for each sample according to the manufacturer’s instructions. Initial processing and normalization of Ct values was performed using Expression Suite release v1.3 (Thermo Fisher Scientific, USA) computer-aided data analysis software. As the expression of none of the miRNAs studied was equally distributed in all samples, global normalization was used, with the calculated average of all miRNAs used as normalizer. Relative expression was calculated using the 2−∆Ct method [22]. The data were converted to relative units and the data obtained were converted on a logarithmic scale. The processed values were used in further statistical analysis.

### 2.5. Statistical Analysis

Variables were tested for normality according to the Shapiro-Wilk test. The results were presented as means ± standard deviation (SD) if the distributions were normal or median and interquartile range (IQR) if the distributions did not meet the criteria of normality. Chi-square (χ^2^) test was used for analysis of data when variables were categorical to analyze the relationship between miRNAs expression and the clinical demographic characteristics. The expression levels of miRNAs among different groups were compared by nonparametric Mann-Whitney U test or Kruskal-Wallis test, as appropriate. Multivariate logistic regression analysis was performed to assess the associations between different miRNAs and the presence of malignancy and metastases. Receiver operating characteristic (ROC) curves were used to determine the accuracy of the miRNA expression studies in the classification of malignant and benign thyroid nodules by evaluating the area under the curve. Youden criterion was used to calculate cut-off values.

Data were analyzed using IBM SPSS Statistics, version 23. A *p* value less than 0.05 was considered as statistically significant. 

## 3. Results

We set out a hypothesis that miRNAs in FNAB can distinguish PTC from benign thyroid nodules and investigated the potential role of miRNAs in the selection of surgical treatment tactics for thyroid nodules with PTC cytology. The distribution of the Bethesda system categories in the PTC and BTN groups showed that category IV was significantly more frequent in the BTN group, while categories V and VI were more frequent in the PTC group (*p* < 0.05). Category III was not significantly different between the BTN and PTC groups (*p* = 0.62). The frequencies of the Bethesda system categories between groups are shown in Table 1.

In addition, we investigated the association between miRNA expression and key clinical demographic parameters to assess the potential utility of miRNAs in predicting PTC invasion and metastasis.

### 3.1. Characteristics of the Study Population

A total of 160 patients were enrolled: 60 in the screening cohort and 100 in the validation cohort. The mean age was similar in both groups, 50.3 (±14.5) and 49.6 (±15.1) years. More than two thirds of the subjects were women.

The median tumor size in pathology was 12 (7.0) mm in the screening group and 16.3 (14.25) mm in the validation group. In both groups, the majority of patients underwent thyroidectomy, 89.6 and 96.6%. In the validation group, 37 patients (86.0%) underwent prophylactic lymphadenectomy of the central neck lymph nodes (CLN), four patients (9.3%) underwent therapeutic lymphadenectomy of the CLN, and two patients (4.7%) underwent a modified lymphadenectomy of central and lateral lymph nodes (LLN). In the validation cohort, 39 patients (73.59%) underwent prophylactic lymphadenectomy of CLN, 11 patients (20.75%) underwent therapeutic lymphadenectomy of CLN, and three patients (5.66%) underwent a modified lymphadenectomy of CLN and LLN. 

The clinical-demographic characteristics of the study population are shown in Table 2.

### 3.2. MiRNA Next-Generation Sequencing Results (Screening Cohort)

Total RNA isolated from thyroid fine needle aspirate material from the first 60 patients was analyzed for miRNA expression profile by next-generation sequencing. To compare the differences in miRNA expression, two analyses were performed: (1) a group of patients (*n* = 33) with PTC without cervical LNM and a group of patients (*n* = 15) with PTC with cervical LNM: (2) a group of patients (*n* = 12) with benign thyroid nodules and a group (*n* = 48) obtained by combining the two PTC groups, PTC with LNM and PTC without LNM. 

When comparing the PTC with LNM and PTC without LNM groups, miRNAs with a mean number of reads greater than 20 in the sample were included in the analysis, with a statistically significant fold change in expression increasing/decreasing >2, *p* < 0.05. There were 12 miRNAs that were found to have a significant difference in fold change.

When comparing PTCs with the group of benign nodules, miRNAs with an average read count greater than 20 in the sample were included in the analysis, with a statistically significant fold change with an increase in expression > 2, *p* < 0.05. There were 12 miRNAs that were found to have a significant difference in fold change.

Of the significantly different miRNAs between PTCs and benign nodules, and between PTCs with LNM and PTCs without LNM, only six miRNAs overlapped. Of these, the three most divergent miRNAs, miR4324, miR200B, miR125A, were selected for further validation. After the literature review, two more miRNAs, miR146B, miR221, which have been mainly described for differentiating both PTC with benign thyroid disease and more aggressive forms of PTC, were selected for further testing [20]. 

### 3.3. Selected miRNA Expression by Real-Time PCR Results (Validation Cohort)

#### 3.3.1. Evaluation of Selected miRNA Relative Expression Differences Based on Clinical Demographic Features of PTC Patients

We performed the analysis of five selected miRNAs’ expression in association with clinical demographic features of PTC. The median and interquartile range (IQR) in the expression of miRNAs are shown in Table 3.

The expression levels of all five miRNAs were similar between patients with microcarcinoma (tumor size ≤10 mm) and those with tumor size >10 mm. 

Significantly higher relative expression of miR221 (*p* = 0.043) was observed in patients with multifocal carcinoma compared with those with solitary carcinoma (Figure 1). No other differences in miRNA expression were found between these groups.

Significantly higher expression of miR125A and miR4324 (*p* < 0.001) was observed in patients with tumor extrathyroidal extension compared with patients without PTC spreading beyond the thyroid capsule. Significant higher expression of miR146B (*p* < 0.001) and miR221 (*p* = 0.027) was found in patient without extrathyroidal tumor extension compared with patients with extrathyroidal PTC extension (Figure 2). No difference in miR200B expression was found between these groups. 

No significant differences in miRNA expression were found among other clinical features, such as patients’ gender, age group, or presence of Hashimoto’s thyroiditis.

#### 3.3.2. Evaluation of Selected miRNA Relative Expression Differences between Benign Thyroid Nodules, PTC with LNM and PTC without LNM Groups

At first we compared the relative expression of the selected five miRNAs between groups of patients (*n* = 36) with BTN and patients (*n* = 64) with PTC. The PTC group was formed by combining the two PTC groups PTC with LNM and PTC without LNM. The expression levels of four miRNAs (miR125A, miR146B, miR221 and miR4324) differed significantly between groups (*p* < 0.05). The expression difference of miR200B was not significant (*p* = 0.135). The median and IQR of the relative expression of all miRNAs are shown in Table 4. 

The results of the miR125A and miR4324 relative expression by RT-PCR confirmed the results of the next-generation sequencing that the relative expression of these miRNAs is higher in the PTC group compared to the benign nodule group. The difference in miR200B expression between the benign and PTC groups as determined by next-generation sequencing was not confirmed by RT-PCR. The difference in expression between the benign and PTC groups for miR146B and miR221 selected from the literature was also confirmed. 

ROC curve analysis was performed to assess the diagnostic accuracy of miR125A, miR146B, miR221 and miR4324, which differ significantly between the groups, in differentiating PTC from benign thyroid nodules. It can be concluded that miR146B (AUC = 0.809; *p* < 0.001; cut-off value—−0.940, Youden—0.660 and miR4324 (AUC = 0.827; *p* < 0.001; cut-off value—0.180, Youden—0.634) were able to identify patients with PTC very well, while miR125A (AUC = 0.716; *p* < 0.001; cut-off value—0.100, Youden—0.538) was able to identify PTC patients well and with very good model quality. The sensitivity and specificity of these three miRNAs are shown in Table 5. MiR221was not evaluated because the model predicts BTN poorly, only 5.6%.

The ROC curves for miR146, miR4324 and miR125A are shown in Figure 3.

Secondly, we analyzed expression levels of selected miRNAs (miR125A, miR146B, miR200B, miR221, miR4324) in PTC without LNM (*n* = 34), PTC with LNM (*n* = 30) and BTN (*n* = 36) groups. The expression levels of four miRNAs (miR125A, miR146B, miR221 and miR4324) differed significantly between groups (*p* < 0.05). The expression difference of miR200B was not significant (*p* = 0.129). Median and interquartile range (IQR) in expression for all miRNAs are shown in Table 6.

Thirdly, we compared the relative expression of miRNAs between groups of patients (*n* = 34) with PTC without LNM and patients (*n* = 30) diagnosed with PTC with LNM. The median and IQR in the expression of all miRNAs are shown in Table 6. Expression analysis of selected miRNAs: miR125A, miR200B and miR4324 between groups by RT-PCR did not confirm the difference in expression between these groups obtained by next-generation sequencing. The difference in expression of the miRNAs (miR146B and miR221) selected from the literature was also not confirmed. Significantly higher expression of miR146B and miR4324 was observed in patients with PTC without LNM compared to patients with PTC with LNM, which is in contrast to the miRNA expression detected in these groups by next-generation sequencing.

Multivariable logistic regression was applied to predict the patient’s histological category of benign thyroid nodules, PTC without LNM or PTC with LNM based on the miRNA. As the logistic regression calculates the ratio of success to failure, the results of the analysis are presented in the form of odds ratios. Two regressors, miR146b and miR4234, were included in the model. The control category was the group of benign thyroid nodules. The model fitted the data well. The correct classification was: PTC without LNM 70.6%, PTC with LNM 56.7%, benign thyroid nodules 72.2%, overall correct classification 67%. Nagelkerke’s pseudo coefficient of determination R^2^ = 0.552. Model likelihood ratio criterion statistic χ^2^ = 67,406, *p* < 0.01. Judging from the likelihood ratio criteria, the regressors are statistically significant.

A 16.0-fold increase in the value of miR146B increases the probability (odds ratio) that the disease will be cancerous without metastasis rather than benign (95% CI [4.188; 61.148]). A 4.2-fold increase in the value of miR146B increases the odds (odds ratio) that the disease will be cancerous with metastasis rather than benign (95% CI [1.692; 10.366]). A 31.9-fold increase in miR4324 value increases the odds (odds ratio) that the disease will be cancerous with metastases rather than benign (95% CI [5.966; 170.765]). A one-fold increase in miR4324 value increases the odds (odds ratio) that the disease will be cancerous with metastasis by 8.3 times more than benign (95% CI [2.155; 31.977]) (Table 7).

## 4. Discussion

PTC is the most common and best prognosis histological variant of the thyroid cancer. However, up to 10% of its cases are progressive, with spread beyond the thyroid gland to lymph nodes or other organs. Recurrence after initial surgical treatment is estimated to be about 20% [23,24]. Therefore, the potentially variable course of PTC poses a dilemma in the choice of the optimal individual treatment. Prognostic indicators of PTC, such as patient’s age, tumor size, tumor extension beyond the thyroid gland, presence of distant metastases and metastases in the neck lymph nodes, postoperative neck ultrasonography, and postoperative thyroglobulin assessment, are used to predict the risk of disease recurrence and mortality [4]. Unfortunately, most of these factors are not known before surgery. Recently, molecular markers for prediction of the outcome of thyroid cancer patients have received particular attention. The inclusion of such molecular markers in risk assessment would allow for the appropriate choice of the right extent of surgery and post-operative treatment. 

Firstly, we performed miRNA profiling of FNAB samples by next-generation sequencing in 60 patients (12 with BTN, 33 with PTC without LNM, 15 with PTC with LNM). We selected three (miR125A, miR200B, miR4324) miRNAs that were the most different between malignant and benign cases, and between the PTC without LNM and PTC with LNM groups, which were further investigated in the remaining patients by RT-PCR. We also selected two more miRNAs (miR146B, miR221) based on the most frequently described miRNAs in the literature, which are associated with papillary carcinoma of the thyroid gland and its more aggressive course. The results showed significantly lower expression of miR146B (*p* < 0.001) and miR221 (*p* = 0.027), and significantly higher expression of miR125A (*p* < 0.001) and miR4324 (*p* < 0.001) in patients with extrathyroidal tumor extension compared to those without extrathyroidal PTC extension. Our results contradict previous data reported in the literature suggesting that miR146B and miR221 are associated with papillary thyroid spread beyond the thyroid capsule [25,26,27,28]. The expression of miR125A and miR4324 in thyroid carcinoma patients has not been described in the literature before. 

We also found a significantly higher relative expression of miR221 (*p* = 0.043) in patients with multifocal carcinoma compared to patients with single foci. This confirms the data reported in the literature [25,27,28,29,30].

We also compared the difference in expression of selected miRNAs between patients with benign thyroid nodules and patients with papillary thyroid carcinoma. To our knowledge, miR125A and miR4324 to differentiate papillary thyroid carcinoma from benign thyroid nodules have not been previously studied and described in the literature. The expression levels, what we found of four miRNAs (miR125A, miR146B, miR221 and miR4324) were significantly different between the groups (*p* < 0.05). The results of RT-PCR for miR125A and miR4324 confirmed the results of next-generation sequencing that the relative expression of these miRNAs is higher in the PTC group compared to the benign nodule group. The difference in miR200B expression between the benign thyroid nodules and PTC groups as determined by next-generation sequencing was not confirmed by RT-PCR. The difference in expression between the benign and PTC groups for the miRNAs miR146B and miR221 selected from the literature was also confirmed. Therefore, our findings of the significantly higher expression of miR146B and miR221 in malignant thyroid nodules confirms the results of studies reported in the literature [26,27,28,31,32,33,34].

Multivariate logistic regression results showed that a one-fold increase in miR146B increased the odds (odds ratio) of being cancerous with no metastases by 16.0-fold and of being cancerous with metastases by 4.2-fold compared to non-malignant disease. In contrast, a one-unit increase in miR4324 increases the odds of having PTC without LNM by 31.9 times and PTC with LNM by 8.3 times compared to benign disease. 

Over the last decade, several studies have been conducted to identify miRNAs associated with PTC metastasis [20]. Chou and co-authors demonstrated that miR146B expression is an independent risk factor for poor prognosis in PTC [35]. Yang and co-authors investigated the expression of miR146B in FNAB samples and found that the expression of this miR146B was significantly higher in patients with neck LNM and more advanced PTC [26]. Several independent groups of investigators have studied miRNAs in histological samples and found that the expression level of miR146B is significantly increased in patients with PTC with metastases [25,36]. However, no significant miR146B expression in PTC with LNM was observed in the other studies [37,38,39].

The miR221/222 are the most frequently studied miRNAs showing an association of their expression change with clinicopathological features of PTC such as neck LNM. A number of studies have found that PTC with LNM has a higher tissue expression of miR221/222 compared to PTC without MTS [25,27,36,39].

In our study, comparing the data from the selected miRNA analyses between groups of patients with papillary thyroid carcinoma with and without metastases in the cervical lymph nodes, we obtained controversial data. The significant difference in the expression of miR125A, miR200B and miR4324 detected by next-generation sequencing was not confirmed by RT-PCR. The expression of miR4324 detected by RT-PCR was significantly higher (*p* = 0.016) in patients with PTC without LNM than in patients with PTC with LNM, whereas the expression of miR200B and miR125A was not significantly different between the groups (*p* > 0.05). Next-generation sequencing data showed that the expression of miR125A, miR146B and miR200B was significantly higher in patients with PTC with LNM than in patients without LNM. Our study did not confirm the difference in expression between the groups selected from the literature [25,26,27,28,29,36,39,40] as the miRNAs most frequently associated with PTC metastasis (miR221 and miR146B). The difference in miR221 expression between the groups was non-significant (*p* = 0.058), whereas miR146B expression was significantly higher (<0.001) in the group of patients without cervical LNM.

What could be the limitations of the study and the reasons why such controversial results were obtained between the groups of patients with PTC with and without LNM? Firstly, we do not know the time which elapsed between tumor onset and diagnosis in each group. As 16 (47.1%) patients in the PTC without LNM group were diagnosed with microcarcinoma, we can assume that there was still time for metastases to occur. Secondly, patients who underwent both prophylactic and therapeutic lymphadenectomy were included. This heterogeneous group of subjects could also have influenced the results. Thirdly, the study included different subtypes of PTC, i.e., classical, oncocytic, solid-trabecular, tall cell, which have different degrees of aggressiveness. The vast majority of the patients included in the study had classical PTC, and the small number of patients with other subtypes of PTC precluded a well-founded subgroup analysis for each disease subtype.

## 5. Conclusions 

Our study showed that the expression analysis of four miRNAs (miR125A, miR221, miR146B and miR4324) could improve the accuracy of fine-needle thyroid node biopsy, which would allow a better differentiation of malignant from benign thyroid nodules. Higher expression of miR125A and miR4324 is associated with papillary carcinoma extending beyond the thyroid gland, while higher expression of miR221 is associated with multifocality, which could help to assess the risk and prognosis of thyroid carcinoma. Analysis of miRNA expression levels and detection of miRNAs in FNAB can be used for the pre-operative diagnosis of thyroid cancer. 

## Figures and Tables

**Figure 1 diagnostics-12-01384-f001:**
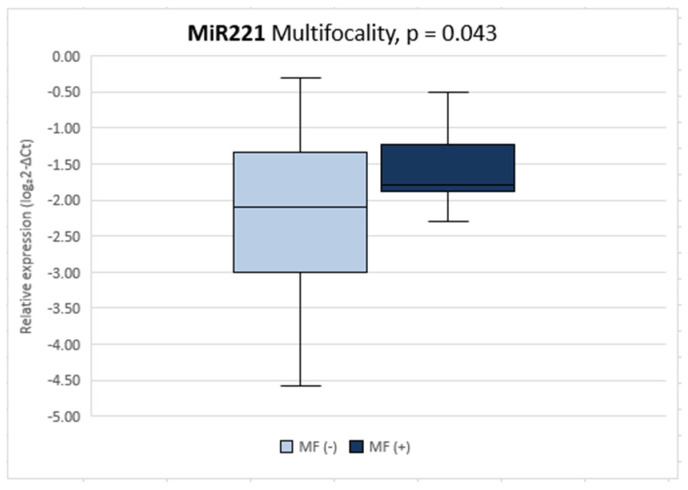
Comparison of miR221 relative expression between solitary and multifocal PTC. The box squares represent the data within 25 and 75 percentiles, the line in the middle shows the median; *p* < 0.05. MF: multifocality.

**Figure 2 diagnostics-12-01384-f002:**
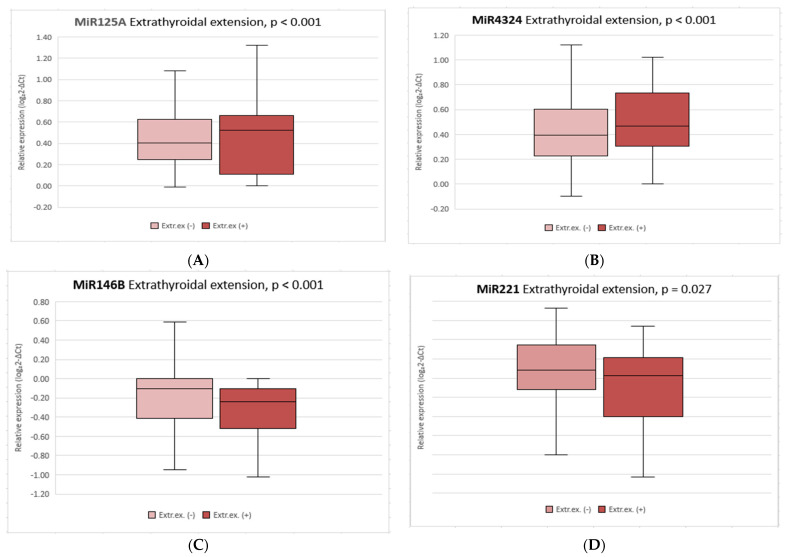
Comparison of miRNAs’ relative expression between PTC with and without extrathyroidal extension; miR125A (**A**); miR4324 (**B**); miR146B (**C**); miR221 (**D**). The box squares represent the data within 25 and 75 percentiles, the line in the middle shows the median; *p* < 0.05.

**Figure 3 diagnostics-12-01384-f003:**
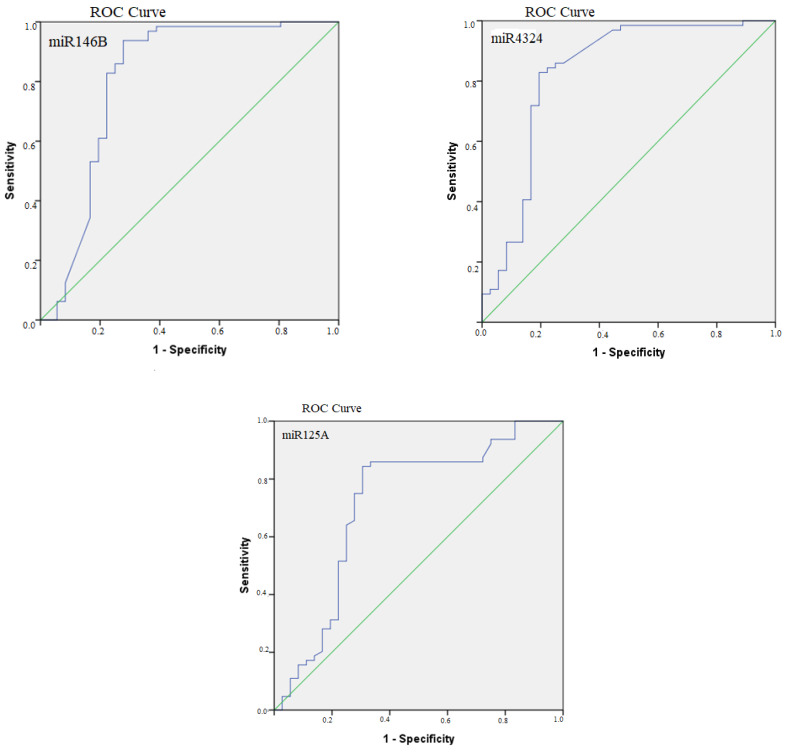
Prognostic value of miRNA in differentiating malignant from benign thyroid nodules. ROC (receiver operating characteristic) curves were used to distinguish the groups.

**Table 1 diagnostics-12-01384-t001:** Distribution of patients by Bethesda categories.

Bethesda Category	BTN (*n*, %)	PTC (*n*, %)	Total (*n*, %)	*p* Value
III	17 (10.6)	20 (12.5)	37 (23.1)	0.62 *
IV	21 (13.1)	3 (1.9)	24 (15.0)	0.001 **
V	6 (3.8)	23 (14.4)	29 (18.1)	0.001 *
VI	4 (2.5)	66 (43.8)	70 (43.8)	<0.001 **
Total (*n*, %)	48 (30.0)	112 (70.0)	160 (100)	

* Pearson’s chi-squared test; ** Fisher’s exact test. BTN—benign thyroid nodes; PTC—papillary thyroid carcinoma.

**Table 2 diagnostics-12-01384-t002:** Clinical-demographic characteristics of the study population.

Characteristic		Screening Cohort	Validation Cohort
All patients			
Number (*n*, %)	Benign:	12 (20.0)	36 (36.0)
	FND	10 (83.3)	23 (63.9)
	FA	2 (16.7)	13 (36.1)
	PTC LNM (−)	33 (55.0)	34 (34.0)
	PTC LNM (+)	15 (25.0)	30 (30.0)
Age (years), mean (SD)		50.3 (14.5)	49.6 (15.1)
Sex (*n*, %)	Male	8 (13.3)	23 (23.0)
	Female	52 (86.7)	77 (77.0)
Tumor size mm, median (IQR)		12 (7.0)	16.5 (14.25)
Patients with PTC			
LNM (*n*, %)	N0	33 (68.75)	34 (53.1)
	N1	15 (31.25)	30 (46.9)
Extrathyroidal extension (*n*, %)	No	26 (54.2)	40 (62.5)
	Yes	22 (45.8)	24 (37.5)
Age groups (*n*, %)	≤55 years	32 (66.7)	46 (71.9)
	>55 years	16 (33.3)	18 (28.1)
Multifocality (*n*, %)	No	31 (64.6)	41 (64.1)
	Yes	17 (35.4)	23 (35.9)
Hashimoto thyroiditis (*n*, %)	No	27 (56.25)	42 (65.6)
	Yes	21 (43.75)	22 (34.4)
Tumor size (mm) (*n*, %)	≤10 mm	25 (52.1)	21 (32.8)
	>10 mm	23 (47.9)	43 (67.2)
T (TNM) (*n*, %)	T1	32 (66.7)	49 (76.5)
	T2	4 (8.3)	9 (14.1)
	T3	12 (25.0)	5 (7,8)
	T4	0 (0.0)	1 (1.6)
Surgery	Lobectomy	5 (10.4)	2 (3.1)
	Thyroidectomy	43 (89.6)	62 (96.9)
Lymphadenectomy	Prophylactic	37 (86.0)	39 (73.59)
	Therapeutic CLN	4 (9.3)	11 (20.75)
	Therapeutic CLN and LLN	2 (4.7)	3 (5.66)
Histological PTC subtype	Classic	43 (89.5)	56 (87.5)
	Infiltrative follicular	3 (6.3)	5 (7.8)
	Oncocytic	1 (2.1)	0 (0.0)
	Solid	0 (0.0)	3 (4.7)
	Tall cell	1 (2.1)	0 (0.0)

FND—follicular nodular disease; FA—follicular adenoma; PTC—papillary thyroid carcinoma; LNM—lymph node metastases; CLN—central lymph nodes; LLN—lateral lymph nodes; SD—Standard deviation; IQR—Interquartile range.

**Table 3 diagnostics-12-01384-t003:** Comparison of miRNA relative expression between clinical-demographic characteristics of PTC.

Clinical Characteristics	*n*	miR125A	miR146B	miR200B	miR221	miR4324
**Tumor size (cm)**						
*≤10* mm	21	0.53 (0.46)	−0.10 (0.42)	−2.35 (1.84)	−1.79 (1.14)	0.49 (0.50)
*>10* mm	43	0.37 (0.43)	−0.21 (0.42)	−2.52 (1.70)	−1.84 (1.62)	0.36 (0.38)
		*p* = 0.183	*p* = 0.424	*p* = 0.426	*p* = 0.424	*p* = 0.183
**MF**						
*No*	40	0.37 (0.49)	−0.23 (0.44)	−2.68 (2.28)	−2.10 (1.77)	0.37 (2.09)
*Yes*	24	0.53 (0.64)	−0.10 (0.29)	−2.35 (0.99)	−1.79 (0.81)	0.58 (0.48)
		*p* = 0.127	*p* = 0.084	*p* = 0.258	*p* = 0.043 *	*p* = 0.107
**Extr.Ex**						
*No*	40	0.40 (0.38)	−0.10 (0.41)	−2.35 (2.16)	−1.79 (1.15)	0.39 (0.40)
*Yes*	24	0.53 (0.64)	−0.24 (0.48)	−2.35 (1.22)	−1.93 (1.63)	0.47 (0.46)
		*p* < 0.001 *	*p* < 0.001 *	*p* = 0.164	*p* = 0.027 *	*p* < 0.001 *
**Cervical LNM**						
*No*	34	0.53 (0.46)	−0.10 (0.20)	−2.35 (1.51)	−1.79 (1.2)	0.58 (0.43)
*Yes*	30	0.37 (0.43)	−0.41 (0.53)	−2.68 (1.71)	−2.30 (1.68)	0.35 (0.31)
		*p* = 0.080	*p* = 0.01 *	*p* = 0.128	*p* = 0.058	*p* = 0.016 *
**Age**						
*≤55 years*	46	0.40 (0.37)	−0.18 (0.41)	−2.35 (1.72)	−1.94 (1.37)	0.37 (0.35)
*>55 years*	18	0.53 (0.86)	−0.10 (0.65)	−2.35 (1.70)	−1.79 (1.11)	0.57 (0.56)
		*p* = 0.931	*p* = 0.996	*p* = 0.867	*p* = 0.268	*p* = 0.278
**HT**						
*No*	42	0.40 (0.47)	−0.14 (0.41)	−2.35 (1.89)	−1.79 (0.99)	0.40 (0.39)
*Yes*	22	0.47 (0.40)	−0.18 (0.52)	−2.35 (1.78)	−1.90 (1.87)	0.48 (0.51)
		*p* = 0.849	*p* = 0.874	*p* = 0.496	*p* = 0.985	*p* = 0.409
**Sex**						
*Male*	12	0.41 (0.36)	−0.21 (0.31)	−2.68 (0.93)	−1.81 (1.34)	0.35 (0.56)
*Female*	52	0.44 (0.46)	−0.10 (0.42)	−2.35 (1.92)	−1.82 (0.72)	0.46 (0.44)
		*p* = 0.705	*p* = 0.509	*p* = 0.286	*p* = 0.874	*p* = 0.161

Mann-Whitney U test, relative expression (log₂2−∆Ct), median and IQR; * *p* < 0.05. MF—multi-focality; Extr.Ex.—extra-thyroidal extension; HT—Hashimoto thyroiditis. LNM—lymph node metastases.

**Table 4 diagnostics-12-01384-t004:** Comparison of miRNA relative expression between BTN and PTC groups.

miRNA	BTN	PTC	*p* Value
*miR125A*	0.02 (0.38)	0.44 (0.43)	<0.001
*miR146B*	−1.49 (1.15)	−0.14 (0.42)	<0.001
*miR200B*	−2.74 (1.55)	−2.35 (1.79)	0.135
*miR221*	−2.34 (1.02)	−1.81 (1.10)	0.032
*miR4324*	−0.12 (0.2)	0.42 (0.40)	<0.001

Mann-Whitney U test, relative expression (log₂2−∆Ct), median and IQR. BTN—benign thyroid nodes, PTC—papillary thyroid carcinoma.

**Table 5 diagnostics-12-01384-t005:** Prognostic value of miRNAs in differentiating malignant from benign nodules.

miRNA	AUC	*p* Value	95% CI	Sensitivity	Specificity
*miR125A*	0.716	<0.001	0.600–0.832	0.667	0.600
*miR146B*	0.809	<0.001	0.700–0.919	0.824	0.885
*miR4324*	0.827	<0.001	0.790–0.926	0.859	0.750

AUC—area under curve; CI—confidence interval. *p* value < 0.001.

**Table 6 diagnostics-12-01384-t006:** Comparison of miRNA relative expression between the BTN, PTC without LNM and PTC with LNM groups; and between PTC with LNM and PTC without LNM groups.

miRNA	BTN	PTC LNM (−)	PTC LNM (+)	*p* Value *	*p* Value **
*miR125A*	0.02 (0.38)	0.53 (0.46)	0.37 (0.43)	0.001	0.080
*miR146B*	−1.49 (1.15)	−0.10 (0.20)	−0.41 (0.53)	<0.001	<0.001
*miR200B*	−2.74 (1.55)	−2.35 (1.51)	−2.68 (1.71)	0.129	0.128
*miR221*	−2.34 (1.02)	−1.79 (1.2)	−2.30 (1.68)	0.024	0.058
*miR4328*	−0.12 (0.2)	0.58 (0.43)	0.35 (0.31)	<0.001	0.016

* Kruskal-Wallis test used to compare the difference between BTN, PTC LNM (−) and PTC LNM (+), relative expression (log₂2−∆Ct), median and IQR. ** Mann-Whitney U test used to compare the difference between PTC LNM (−) and PTC LNM (+); relative expression (log₂2−∆Ct), median and IQR. BTN—benign thyroid nodes; PTC—papillary thyroid carcinoma; LNM—lymph nodes metastases.

**Table 7 diagnostics-12-01384-t007:** Multivariable logistic regression analysis predicting histological category of PTC without LNM or PTC with LNM. The control category—BTN group.

PTC	miRNA	OR (95% CI)	*p* Value
PTC LNM (−)	miR146B miR4324	16.0 (4.188–61.148) 31.9 (5.966–170.765)	<0.001 <0.001
PTC LNM (+)	miR146B miR4324	4.2 (1.692–10.366) 8.3 (2.155–31.977)	0.002 0.002

PTC: papillary thyroid carcinoma; LNM: lymph nodes metastases; OR: odds ratio; CI: confidence interval.

## Data Availability

Not applicable.

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
