# Peer review of "Diagnostic Significance of FNAB miRNA Expression in Papillary Thyroid Carcinoma"

_diagnostics, 2022, doi:10.3390/diagnostics12061384_

Round 1

Reviewer 1 Report

Overall, the idea of the work is interesting. The findings of the work could have merit in the related field. The study limitations have been mentioned by the end of the discussion. Just some concerns should be addressed.

 Abstract

- Lines 16 and 17: The aim of categorizing the microRNAs into two subgroups was not clear for the readers, in particular before complete reading of the manuscript.

- Lines 19-26: If the authors have no specific aim in arranging the miRNAs in their elaboration, please keep consistency throughout the text in ordering them according to their mention first time in this context.

 Introduction

Generally, the references related to statistics are better to be recent ones, please update Ref. (1) as it was since 2008.

- Lines 53 and 54: “Analysis of miRNAs expression in FNAB samples may improve perioperative decision making for patients with PTC”. Some detailed elaborations with miRNA examples from previous literature are required. What are the advantages of this type of non-coding RNAs to be applied as diagnostic/prognostic assays in the patients with cancer, in particular in such type of sampling?

Methods

- Line  67: “The Bethesda system” The authors should provide a supportive reference for this system. It is not provided that all future readers will be specialty-related ones.

- The exclusion criteria of the participants was not clear.

- Line 77” Following the literature review, two more miRNAs – miR146B, miR221” the authors should provide appropriate citations to support this sentence.

- Lines 85-86: “according to an approved diagnostic protocol”. What is this protocol? please, provide a supportive citation if will not detailed.

- Line 109: “RL-PCR” please revise.

- Line 134: The authors should revise the citation of the SPSS applied for their analysis as version 28 was released from IBM Corp.

Line 227: the authors should revise “miR146B (B); miR221 (C); miR4324 (D)” to match the figure panels.

 - Could the authors provide a supportive figure for the Multivariable logistic regression analysis to facilitate following them in their explanation?

 Tables

As each table should be self-explanatory, the authors should ensure that each Table footer includes the type of data presentation, the statistical test applied, the level of significance, in addition to the abbreviations (mentioned).

Reviewer 2 Report

Dear Authors,

This is a well and clearly written paper. I commend you for your good work. I have few minor comments:

1.       Please, add some references into the 2nd paragraph  of the Introduction part (pg 1,2, line 41-50)

2.       I suggest you to add the row in the Table 2 with the post-operative final diagnosis of BTN (Follicular adenomas, Hürthle adenomas…) and PTC subtypes of both, the screening and validation cohort

3.       In the 3.3.2. section, I suggest you to firstly compare BTN with total PTC, and afterwards BTN, PTC lnm- and PTC lnm+ . Even more, it would be more convenient to merge the Table4 with the Table 7, by adding one column in Table 4 with the p-value results from the Table 7 (the merged table would have two p-value column , the 1st one with the p of BTN, PTC lnm- and PTC lnm+ comparison, and the 2nd with the p of PTC lnm- and PTC lnm+ comparison)

4.       The results would be more complete with the miR-221 ROC curve presented in Fig 4.

5.       Please, insert the cut-off values for miR-125A, -146B and -4324, based on ROC analysis (in the text or in the Table 6)

6.       Please, rewrite the paragraph in the results with (present) Table 7 explanations (pg 11-12), because written in this way, I found it confusing. I suggest you to firstly explain the results shown in (present) Tab7 (with all significances), and then compare the results gained with RT-PCR with the results gained by NGS. Afterwards, you may mention the disagreement and describe what the difference was. Additionally, as I could noticed, miR146B expression was not tested by NGS (pg 12 and pg13,ln59)

7.       Pg13, ln24 – Please add how your results contradict previous data reported (did the previous data suggested no change or higher miR-156b,-221 levels in FNAB of PTC lnm+ cases compared to PTC lnm- or compared to BTN?)

8.       Typing mistakes: in Tabs 4,5 and 7 it is written miR4328 instead of miR4324; and on pg11,ln71 RL-PCR instead of RT-PCR
